# Representation compression and generalization in Deep Neural Networks

## Abstract

Understanding the groundbreaking performance of Deep Neural Networks (DNNs) is one of the greatest challenges to the scientific community today. In this work we introduce an information theoretic viewpoint on the behavior of deep networks optimization processes and their generalization abilities. Specifically, we study DNNs on the Information Plane, the plane of the mutual information between each layer with and the input variable, and with the desired label, during the training dynamics. We show that the training of the network is characterized by a rapid increase in the mutual information (MI) between the layers and the target label, followed by a longer decrease in the MI between the layers and the input variable.vFurther, we explicitly show that these two fundamental information-theoretic quantities govern the generalization error of the network, by introducing a new generalization-gap bound that is exponential in the input representation compression. The analysis focuses on typical patterns of large-scale problems. For this purpose, we introduce a novel analytic bound on the mutual information between consecutive layers in the network. An important consequence of our analysis is a super-linear boost in training time with the number of non-degenerate hidden layers, demonstrating the computational benefit of the hidden layers.

## 1 Introduction

Deep Neural Networks (DNNs) heralded a new era in predictive modeling and machine learning. Their ability to learn and generalize has set a new bar on performance, compared to state-of-the-art methods. This improvement is evident across almost every application domain, and especially in areas that involve complicated dependencies between the input variable and the target label (Le-Cun et al., 2015). However, despite their great empirical success, there is still no comprehensive understanding of their optimization process and its relationship to their (remarkable) generalization abilities.

This work examines DNNs from an information-theoretic viewpoint. For this purpose we utilize the Information Bottleneck principle (Tishby et al., 1999). The Information Bottleneck (IB) is a computational framework for extracting the most compact, yet informative, representation of the input variable ($X$), with respect to a target label variable ($Y$). The IB bound defines the optimal trade-off between representation complexity and its predictive power. Specifically, it is achieved by minimizing the mutual information (MI) between the representation and the input, subject to the level of MI between the representation and the target label.

Recent results (Shwartz-Ziv & Tishby, 2017), demonstrated that the layers of DNNs tend to converge to the IB optimal bound. The results pointed to a distinction between two phases of the training process. The first phase is characterized by an increase in the MI with the label (i.e. fitting the training data), whereas in the second and most important phase, the training error was slowly reduced with a decrease in mutual information between the layers and the input (i.e. representation compression). These two phases appear to correspond to fast convergence to a flat minimum (drift) following a random walk, or diffusion, in the vicinity of the training error's flat minimum, as reported in other studies (e.g. (Zhang et al., 2018a)).

These observations raised several interesting questions: (a) which properties of the SGD optimization cause these two training phases? (b) how can the diffusion phase improve generalization perfor-

mance? (c) can the representation compression explain the convergence of the layers to the optimal IB bound? (d) can this diffusion phase explain the benefit of many hidden layers?

In this work we attempt to answer these questions. Specifically, we draw important connections between recent results inspired by statistical mechanics and information-theoretic principles. We show that the layers of a DNN indeed follow the behavior described by Shwartz-Ziv & Tishby (2017). We claim that the reason may be found in the Stochastic Gradient Decent (SGD) optimization mechanism. We show that the first phase of the SGD is characterized by a rapid decrease in the training error, which corresponds to an increase in the MI with the labels. Then, the SGD behaves like non-homogeneous Brownian motion in the weights space, in the proximity of a flat error minimum. This non-homogeneous diffusion corresponds to a decrease in MI between the layers and the input variable, in "directions" that are irrelevant to the target label.

One of the main challenges in applying information theoretic measures to real-world data is a reasonable estimation of high dimensional joint distributions. This problem has been extensively studied over the years (e.g. (Paninski, 2003)), and has led the conclusion that there is no "efficient" solution when the dimension of the problem is large. Recently, a number of studies have focused on calculating the MI in DNNs using Statistical Mechanics. These methods have generated promising results in a variety of special cases (Gabrié et al., 2018), which support many of the observations made by Shwartz-Ziv & Tishby (2017).

In this work we provide an analytic bound on the MI between consecutive layers, which is valid for any non-linearity of the units, and directly demonstrates the compression of the representation during the diffusion phase. Specifically, we derive a Gaussian bound that only depends on the linear part of the layers. This bound gives a super linear dependence of the convergence time of the layers, which in turn enables us to prove the super-linear computational benefit of the hidden layers. Further, the Gaussian bound allows us to study mutual information values in DNNs in real-world data without estimating them directly.

## 1.1 PRELIMINARIES AND NOTATIONS

Let $X \in \mathcal{X}$ and $Y \in \mathcal{Y}$ be a pair of random variables of the input patterns and their target label (respectively). Throughout this work, we consider the practical setting where $X$ and $Y$ are continuous random variables that are represented in a finite precision machine. This means that both $X$ and $Y$ are practically binned (quantized) into a finite number of discrete values. Alternatively, $X, Y$ may be considered as continuous random variables that are measured in the presence of small independent additive (Gaussian) noise, corresponding to their numerical precision. We use these two interpretations interchangeably, at the limit of infinite precision, where the limit is applied at the final stage of our analysis.

We denote the joint probability of $X$ and $Y$ as $p(x, y)$, whereas their corresponding MI is defined as $I(X; Y) = D\left[p(y|x)||p(y)\right] = D\left[p(x|y)||p(x)\right]$. We use the standard notation $D[p||q]$ for the Kullback-Liebler (KL) divergence between the probability distributions $p$ and $q$. Let $f_{W^K}(x)$ denote a DNN, with $K$ hidden layers, where each layer consists of $d_k$ neurons, each with some activation function $\sigma_k(x)$, for $k = 1, \ldots, K$. We denote the values of the $k^{th}$ layer by the ranom vector $T_k$. The DNN mapping between two consecutive layers is defined as $T_k = \sigma_k\left(W_k T_{k-1}\right)$, where $W_k$ is a $d_k \times d_{k-1}$ real weight matrix. Note that we consider both the weights, $W_k$ and the layer representations, $T_k$, as stochastic entities, because they depend on the stochastic training rule of the network and the random input pattern (as described in Section 2.1). However, when the network weights are given, the weights are fixed realizations of the random training process (i.e. they are "quenched"). Note that given the weights, the layers form a Markov chain of successive internal representations of the input variable $X$: $Y \to X \to T_1 \to \ldots \to T_K$, and their MI values obey a chain of Data Processing Inequalities (DPI), as discussed by Shwartz-Ziv & Tishby (2017).

We denote the set of all $K$ layers weight matrices as $W^K = \{W_1, \ldots, W_K\}$. Let the *training sample*, $S^n = \{(x_1, y_1), \ldots, (x_n, y_n)\}$ be a collection of $n$ independent samples from $p(x, y)$. Let $\ell_{W^K}(x_i, y_i)$ be a (differentiable) loss function that measures the discrepancy between a prediction of the network $f_{W^K}(x_i)$ and the corresponding true target value $y_i$, for a given set of weights $W^K$. Then, the empirical error is defined as $\mathcal{L}_{W^K}(S^n) = \frac{1}{n}\sum_{i=1}^{n}\ell_{W^K}(x_i, y_i)$. The corresponding error gradients (with respect to the weights) are denoted as $\nabla_{W^K}\mathcal{L}_{W^K}(S^n)$.

## 2 Deep Neural Networks

### 2.1 Training the Network – the SGD Algorithm

Training a DNN corresponds to the process of setting the values of weights $W^K$ from a given set of samples $S^n$. This is typically done by minimizing the empirical error, which approximates the expected loss. The SGD algorithm is a common optimization method for this purpose (Robbins & Monro, 1951).

Let $S^{(m)}$ be a random set of $m$ samples drawn (uniformly, with replacement) from $S^n$, where $m < n$. We refer to $S^{(m)}$ as a *mini-batch* of $S^n$. Define the corresponding empirical error and gradient of the mini-batch as $\mathcal{L}_{W^K}\left(S^{(m)}\right) = \frac{1}{m}\sum_{\{x_i,y_i\}\in S^{(m)}}\ell_{W^K}(x_i,y_i)$ and $\nabla_{W^K}\mathcal{L}_{W^K}\left(S^{(m)}\right) = \frac{1}{m}\sum_{\{x_i,y_i\}\in S^{(m)}}\nabla_{W^K}\ell_{W^K}(x_i,y_i)$ respectively. Then, the SGD algorithm is defined by the update rule: $W^K(l) = W^K(l-1) - \eta\nabla_{W^K(l-1)}\mathcal{L}_{W^K(l-1)}\left(S^{(m)}\right)$, where $W^K(l)$ are the weights after $l$ iterations of the SGD algorithm and $\eta \in \mathbb{R}_+$ is the learning rate.

### 2.2 The Different Phases of SGD Optimization

The SGD algorithm plays a key role in the astonishing performance of DNNs. As a result, it has been extensively studied in recent years, especially in the context of flexibility and generalization (Chee & Toulis, 2017). Here, we examine the SGD as a stochastic process, that can be decomposed into two separate phases. This idea has been studied in several works (Murata, 1998; Jin et al., 2017; Hardt et al., 2015). Murata argued that stochastic iterative procedures are initiated at some starting state and then move through a fast *transient phase* towards a *stationary phase*, where the distribution of the weights becomes time-independent. However, this may not be the case when the SGD induces non-isotropic state dependent noise, as argued, for example, by Chaudhari & Soatto (2017).

In contrast, Shwartz-Ziv & Tishby (2017) described the transient phase of the SGD as having two very distinct dynamic phases. The first is a *drift* phase, where the means of the error gradients in every layer are large compared to their batch-to-batch fluctuations. This behaviour is indicative of small variations in the gradient directions, or *high-SNR gradients*. In the second part of the transient phase, which they refer to as *diffusion*, the gradient means become significantly smaller than their batch-to-batch fluctuations, or *low-SNR gradients*. The transition between the two phases occurs when the training error saturates and weights growth is dominated by the gradient batch-to-batch fluctuations. Typically, most SGD updates are expended in the diffusion phase before reaching Murata's stationary phase. In this work we rigorously argue that this diffusion phase causes the representation compression; the observed reduction in $I(T_k; X)$, for most hidden layers.

### 2.3 Drift and Diffusion with SGD

It is well known that the discrete time SGD (2.1) can be considered as an approximation of a continuous time stochastic gradient flow if the discrete-time iteration parameter $l$ is replaced by a continuous parameter $\tau$. Li et al. (2015) recently showed that when the mini-batch gradients are unbiased with bounded variance, the discrete-time SGD is an approximation of a continuous-time Langevin dynamics,

$$dW^K(\tau) = -\nabla_{W^K(\tau)}\mathcal{L}_{W^K(\tau)}(S_n)\,d\tau + \sqrt{2\beta^{-1}C\left(W^K(\tau)\right)}dB(\tau) \tag{1}$$

where $C\left(W^K(\tau)\right)$ is the sample covariance matrix of the weights, $B(\tau)$ is a standard Brownian motion (Wiener process) and $\beta$ is the Langevin temperature constant. The first term in (1) is called the gradient flow or drift component, and the second term corresponds to random diffusion. Although, this stochastic dynamics hold for the entire SGD training process, the first term dominates the process during the high SNR gradient phase, while the second term becomes dominant when the gradients are small, due to saturation of the training error in the low SNR gradient phase. Hence, these two SGD phases are referred to as drift and diffusion.

The *mean $L_2$ displacement* (MSD) measures the Euclidean distance from a reference position over time, which is used to characterize a diffusion process. Normal diffusion processes are known to exhibit a power-law MSD in time, $\mathbb{E}\left[\left\|W^K(\tau) - W^K(0)\right\|_2\right] \sim \gamma t^\alpha$, where $t$ is the diffusion time,

$\gamma$ is related to the diffusion coefficient, and $0 < \alpha \le 0.5$ is the diffusion exponent. For a standard flat space diffusion, the MSD increases as a square root of time ($\alpha = 0.5$). Hu et al. (2017) showed (empirically) that the weights' MSD, in a DNNs trained with SGD, indeed behaves (asymptotically) like a normal diffusion, where the diffusion coefficient $\gamma$ depends on the batch size and learning rate. In contrast, Hoffer et al. (2017) showed that the weights' MSD demonstrates a much slower logarithmic increase. This type of dynamics is also called "ultra-slow" diffusion.

## 3 INFORMATION PLANE ANALYSIS

Following Tishby & Zaslavsky (2015) and Shwartz-Ziv & Tishby (2017), we study the layer representation dynamics in the two-dimensional $(I(X; T_k), I(T_k; Y))$ plane. Specifically, for any input and target variables, $X, Y$, let $T \triangleq T(X)$ denote a representation, or an encoding (not necessarily deterministic), of $X$. Clearly, $T$ is fully characterized by its *encoder*, the conditional distribution $p(t|x)$. Similarly, let $p(y|t)$ denote any (possibly stochastic) *decoder* of $Y$ from $T$. Given a joint probability function $p(x, y)$, the *Information Plane* is defined the set of all possible pairs $I(X; T)$ and $I(T; Y)$ for any possible representation, $p(T|X)$.

It is evident that not all points on the plane are feasible (achievable), as there is clearly a tradeoff between these quantities; the more we compress $X$ (reduce $I(X; T)$), the less information can be maintained about the target, $I(T; Y)$.

Our analysis is based on the fundamental role of these two MI quantities. We argue that for large scale (high dimensional $X$) learning, for almost all (*typical*) input patterns, with mild assumptions (ergodic Markovian input patterns): (i) the MI values concentrate with the input dimension; (ii) the minimal sample complexity for a given generalization gap is controlled by $I(X; T)$; and (iii) the accuracy - the generalization error - is governed by $I(T; Y)$, with the Bayes optimal decoder representation.

Here, we argue that the sample-size - accuracy trade-off, of all large scale representation learning, is characterized by these two MI quantities. For DNNs, this amounts to a dramatic reduction in the complexity of the analysis of the problem. We discuss these ideas in the following sections and prove the connection between the input representation compression $I(T; X)$, the generalization gap (the difference between training and generalization errors), and the minimal sample complexity (Theorem 1 below).

### 3.1 LABEL INFORMATION AND GENERALIZATION ERROR

Optimizing mutual information quantities is by no means new in either supervised and unsupervised learning (Deco & Obradovic, 2012; Linsker, 1988; Painsky et al., 2016). This is not surprising, as it can be shown that $I(T; Y)$ corresponds to the irreducible error when minimizing the logarithmic loss (Painsky & Wornell, 2018b; Harremoes & Tishby, 2007). Here, we emphasize that $I(T; Y)$, for the optimal decoder of the representation $T$, governs all reasonable generalization errors (under the mild assumption that label $y$ is not completely deterministic; $p(y|x)$ is in the interior of the simplex, $\Delta(Y)$, for all typical $x \in X$). First, note that with the Markov chain $Y - X - T$, $I(T; Y) = I(X; Y) - \mathbb{E}_{X,T} D\left[p(y|x)||p(y|t)\right]$. By using the Pinsker inequality (Cover & Thomas, 2012) the variation distance between the optimal and the representation decoders can be bound by their KL divergence,

$$D\left[p(y|x)||p(y|t)\right] \ge \frac{1}{2\ln 2}\left|p(y|x) - p(y|t)\right|_1^2. \tag{2}$$

Hence, by maximizing $I(T; Y)$ we minimize the expected *variation risk* between the representation decoder $p(y|t)$ and $p(y|x)$. For more similar bounds on the error measures see (Painsky & Wornell, 2018a).

### 3.2 REPRESENTATION COMPRESSION AND SAMPLE COMPLEXITY

The *Minimum Description Length* (MDL) principle (Rissanen, 1978) suggests that the best representation for a given set of data is the one that leads to the minimal code-length needed to represent of the data. This idea has inspired the use of $I(X; T)$ as a regularization term in many learning

problems (e.g. Chigirev & Bialek (2004); Painsky et al. (2018)). Here, we argue that $I(X;T)$ plays a much more fundamental role; we show that for large scale (high dimensional $X$) learning and for typical input patterns, $I(X;T)$ controls the sample complexity of the problem, given a generalization error gap.

**Theorem 1** (Input Compression bound). *Let $X$ be a $d$-dimensional random variable that obeys an ergodic Markov random field probability distribution, asymptotically in $d$. Let $T \triangleq T(X)$ be a representation of $X$ and denote by $T_m = \{(t_1, y_1), \ldots, (t_m, y_m)\}$ an $m$-sample vector of $T$ and $Y$, generated with $m$ independent samples of $x_i$, with $p(y|x_i)$ and $p(t|x_i)$. Assume that $p(x, y)$ is bounded away from 0 and 1 (strictly inside the simplex interior). Then, for large enough $d$, with probability $1 - \delta$, the typical expected squared generalization gap satisfies*

$$\left| \mathcal{L}(T_m) - \mathbb{E}_{T_m} \left[ \mathcal{L}(T_m) \right] \right|^2 \leq \frac{2^{I(X;T)} + \log \frac{2}{\delta}}{2m}. \tag{3}$$

*where the typicality follows the standard Asymphotic Equipartition Property (AEP) (Cover & Thomas, 2012).*

A proof of this Theorem is given in Appendix A. This Theorem is also related to the bound proved by Shamir et al. (2010), with the typical representation cardinality, $|T(X)| \approx 2^{I(T;X)}$. The ergodic Markovian assumption is common in many large scale learning problems. It means that $p(x) \approx \prod_{i=1:d} p(x_i|Pa(x_i))$, where $Pa(x_i)$ is a finite set of adjacent "parents" of $x_i$ in the $d$ dimensional pattern $X$.

The consequences of this input-compression bound are quite striking: the generalization error decreases exponentially with $I(X;T)$, once $I(T;X)$ becomes smaller than $\log 2m$ - the query sample-complexity. Moreover, it means that $M$ bits of representation compression, beyond $\log 2m$, are equivalent to a factor of $2^M$ training examples. The tightest bound on the generalization bound is obtained for the most compressed representation, or the last hidden layer of the DNN. The input-compression bound can yield a tighter and more realistic sample complexity than any of the worst-case PAC bounds with any reasonable estimate of the DNN class dimensionality, as typically the final hidden layers are compressed to a few bits.

Nevertheless, two important caveats are in order. First, the layer representation in Deep Learning are learned from the training data; hence, the encoder, the partition of the typical patterns $X$, and the *effective "hypothesis class"*, depend on the training data. This can lead to considerable over-fitting. Training with SGD avoids this potential over-fitting because of the way the diffusion phase works. Second, for low $I(T;Y)$ there are exponentially (in $d$) many random encoders (or soft partitions of $X$) with the same value of $I(T;X)$. This seems to suggest that there is a missing exponential factor in our estimate of the hypothesis class cardinality. However, note that the vast majority (almost all) of these possible encoders are never encountered during typical SGD optimization. In other words, they act like a "dark hypothesis space" which is never observed and does not affect the generalization bound. Moreover, as $I(T;Y)$ increases, the number of such random encoders rapidly collapses all the way to $O(1)$ when $I(T;Y)$ approaches the optimal IB limit, as we show next.

### 3.3 The Information Bottleneck limit

As presented above, we are interested in the boundary of the achievable region in the information plane, or in encoder-decoder pairs that minimize the sample complexity (minimize $I(X;T)$) and generalize well (maximize $I(T;Y)$).

These optimal encoder-decoder pairs are given precisely by the Information Bottleneck framework (Tishby et al., 1999), which is formulated by the following optimization problem: $\min_{p(t|x)} I(X;T) - \beta I(T;Y)$, over all possible encoders-decoders pairs that satisfy the Markov condition $Y - X - T$. Here $\beta$ is a positive Lagrange multiplier associated with the decoder information on $I(T;Y)$, which also determines the complexity of the representation.

The Information Bottleneck limit defines the set of optimal encoder-decoder pairs, for the joint distribution $p(x, y)$. Furthermore, it characterizes the achievable region in the Information Plane, similar to Shannon's Rate Distortion Theory (RDT) (Cover & Thomas, 2012). By our previous analysis it also determines the optimal tradeoff between sample complexity and generalization error. The IB can only be solved analytically in very special cases (e.g., jointly Gaussian $X, Y$ (Chechik

et al., 2005)). In general, a (locally optimal) solution can be found by iterating the self-consistent equations, similar to the Arimoto- Blahut algorithm in RDT (Tishby et al., 1999). For general distributions, no efficient algorithm for solving the IB is known, though there are several approximation schemes (Chalk et al., 2016; Painsky & Tishby, 2017). The self-consistent equations are exactly satisfied along the IB limit, aka *the Information Curve*.

# 4 THE INFORMATION PLANE AND SGD DYNAMICS FOR DNNS

By applying the DPI to the Markov chain of the DNN layers we obtain the following chains: $I(X;T_1) \geq I(X;T_2) \geq \cdots \geq I(X;T_k) \geq I(X;\hat{Y})$ and $I(X;Y) \geq I(T_1;Y) \geq \cdots \geq I(T_k;Y) \geq I(\hat{Y};Y)$ where $\hat{Y}$ is the output of the network. The pairs $(I(X;T_k), I(T_k,Y))$, for each SGD update, form a unique concentrated *Information Path* for each layer of a DNN, as demonstrated by Shwartz-Ziv & Tishby (2017).

For any fixed realization of the weights, the network is, in principle, a deterministic map. This does not imply that information is not lost between the layers; the inherent finite precision of the layers, with possible saturation of the nonlinear activation functions $\sigma_k$, can result in non-invariable mapping between the layers. Moreover, we argue below that for large networks this mapping becomes effectively stochastic due to the diffusion phase of the SGD.

On the other hand, the Information Plane layer paths are invariant to invertible transformations of the representations $T_k$. Thus the same paths are shared by very different weights and architectures, and possibly different encoder-decoder pairs. This freedom is drastically reduced when the target information, $I(T_k,Y)$, increases and the layers approach the IB limit. Minimizing the training error (ERM), together with standard uniform convergence arguments clearly increase $I(T;Y)$, but what in the SGD dynamics can lead to the observed representation compression which further improves generalization? Moreover, can the SGD dynamics push the layer representations all the way to the IB limit, as claimed in Shwartz-Ziv & Tishby (2017)?

We provide affirmative answers to both questions, using the properties of the drift and diffusion phases of the SGD dynamics.

## 4.1 REPRESENTATION COMPRESSION BY DIFFUSION

In this section we quantify the roles of the drift and diffusion SGD phases and their influence on the MI between consecutive layers. Specifically, we show that the drift phase corresponds to an increase in information with the target label $I(T_k;Y)$, whereas the diffusion phase corresponds to representation compression, or reduction of the $I(X;T_k)$. The representation compression is accompanied by further improvement in the generalization.

The general idea is as follows: the drift phase increases $I(T_k;Y)$ as it reduces the cross-entropy empirical error. On the other hand, the diffusion phase in high dimensional weight space effectively adds an independent non-uniform random component to the weights, mostly in the directions that do not influence the loss - i.e, *irrelevant directions*. This results in a reduction of the SNR of the irrelevant features of the patterns, which leads to a reduction in $I(X;T_k)$, or representation compression. We further argue that different layers filter out different irrelevant features, resulting in their convergence to different locations on the Information Plane.

## 4.2 THE SGD COMPRESSION MECHANISM

First, we notice that the DPI implies that $I(X;T_k) \leq I(T_{k-1};T_k)$. We focus on the second term during the diffusion phase and prove an asymptotic upper bound for $I(T_{k-1};T_k)$, which reduces sub-linearly with the number of SGD updates. For clarity, we describe the case where $T_k \in \mathbb{R}^{d_k}$ is a vector and $T_{k+1} \in \mathbb{R}$ is a scalar. The generalization to higher $d_{k+1}$ is straightforward. We examine the network during the diffusion phase, after $\tau$ iterations of the SGD beyond the drift-diffusion transition. For each layer, $k$, the weights matrix, $W^k(\tau)$ can be decomposed as follows,

$$W^k(\tau) = W^{k\star} + \delta W^k(\tau). \tag{4}$$

The first term, $W^{k\star}$, denotes the weights at the end of the drift phase ($\tau_0 = 0$) and remains constant with increasing $\tau$. As we assume that the weights converge to a (local, flat) optimum during the

drift phase, $W^{k^\star}$ is close to the weights at this local optimum. The second term, $\delta W^k(\tau)$, is the accumulated Brownian motion in $\tau$ steps due to the batch-to-batch fluctuations of the gradients near the optimum. For large $\tau$ we know that $\delta W^k(\tau) \sim \mathcal{N}(0, \tau C(W^k(\tau_0)))$ where $\tau_0$ is the time that the diffusion phase began. Note that for any given $\tau$, we can treat the weights as a fixed (quenched) realization, $w^k(\tau)$, of the random Brownian process $W^k(\tau)$. We can now model the mapping between the layers $T_k$ and $T_{k+1}$ at that time as

$$T_{k+1} = \sigma_k \left( w^{*^T} T_k + \delta w^k(\tau)^T T_k + Z \right) \tag{5}$$

where $w^* \in \mathbb{R}^{d_k}$ is the SGD's empirical minimizer, and $\delta w \in \mathbb{R}^{d_k}$ is a realization from a Gaussian vector $\delta w \sim \mathcal{N}(0, C_{\delta w})$, of the Brownian process discussed in Section 2.3. In addition, we consider $Z \sim \mathcal{N}(0, \sigma_z^2)$ to be the small Gaussian measurement noise, or quantization, independent of $\delta w^k$ and $T_k$. This standard additive noise allows us to treat all the random variables as continuous.

For simplicity we first assume that the $d_k$ components of $T_k$ have zero mean and are asymptotically independent for $d_k \to \infty$.

**Proposition 2.** *Under mild technical conditions which are met with probability $1$ in standard deep learning (see Appendix B), we have that*

$$\frac{1}{\sqrt{\sigma_{T_k}^2}} \left[ \frac{w^{*^T} T_k}{||w^*||_2} \quad \frac{\delta w^T T_k}{||\delta w||_2} \right]^T \xrightarrow[d_k \to \infty]{\mathcal{D}} \mathcal{N}(0, I) \tag{6}$$

*almost surely, where $\sigma_{T_k}^2$ is the variance of the components of $T_k$.*

A proof for this CLT proposition is given in Appendix B.

Proposition 2 shows that under standard conditions, $w^{*^T} T_k$ and $\delta w^T T_k$ are asymptotically jointly Gaussian and independent, almost surely. We stress that the components of $T_k$ do not have to be identically distributed to satisfy this property; Proposition 2 may be adjusted for this case with different normalization factors. Similarly, the independence assumption on $T_k$ can easily be relaxed to Markovian ergodic, as we assume the input patterns. Finally, it is easy to verify that Proposition 2 can be extended to the general case where $w^*, \delta w \in \mathbb{R}^{d_k \times d_{k+1}}$, under similar technical conditions.

We can now bound the mutual information between $T_{k+1}$ and the linear projection of the previous layer $W^* T_k$, during the diffusion phase, for sufficiently high dimensions $d_k, d_{k+1}$, under the conditions above. Note that in this case, (5) behaves like an additive Gaussian channel where $w^{*^T} T_k$ is the signal and $\delta w^T T_k + Z$ is an independent additive Gaussian noise (i.e., independent of signal and normally distributed). Hence, for sufficiently large $d_k$ and $d_{k+1}$, we have

$$I(T_{k+1}; T_k | w^*) \leq I(T_{k+1}; w^{*^T} T_k | w^*) \leq I\left( w^{*^T} T_k + \delta w^T T_k + Z; w^{*^T} T_k | w^* \right) = \tag{7}$$

$$\frac{1}{2} \log \left( \frac{\left| \sigma_{T_k}^2 w^{*^T} w^* + \sigma_{T_k}^2 \delta w^T \delta w + \sigma_z^2 I \right|}{\left| \sigma_{T_k}^2 \delta w^T \delta w + \sigma_z^2 I \right|} \right)$$

almost surely, where the first inequality is due to DPI for the Markov chain $T_k - w^{*^T} T_k - T_{k+1}$. Finally, we apply an orthogonal eigenvalue decomposition to this multivariate Gaussian channel (7). Let $\delta w^T \delta w = Q \Lambda Q^T$ where $QQ^T = I$ and $\Lambda$ is a diagonal matrix whose diagonal elements are the corresponding eigenvalues, $\lambda_i$, of $\delta w^T \delta w$. Then, we have that

$$\left| \sigma_{T_k}^2 w^{*^T} w^* + \sigma_{T_k}^2 \delta w^T \delta w + Z \right| = \sigma_{T_k}^2 |Q| \cdot |Q^T w^{*^T} w^* Q + \Lambda + \frac{\sigma_z^2}{\sigma_{T_k}^2} Q^T Q| \cdot |Q^T| = \tag{8}$$

$$\sigma_{T_k}^2 |Q^T w^{*^T} w^* Q + \Lambda + \frac{\sigma_z^2}{\sigma_{T_k}^2} I| \leq \sigma_{T_k}^2 \prod_{i=1}^{d_{k+1}} \left( A_{ii} + \lambda_i + \frac{\sigma_z^2}{\sigma_{T_k}^2} \right)$$

where $A \triangleq Q^T W^{*T} W^* Q$. The last inequality is due to the Hadamard inequality. Plugging (8) into (7) yields that for sufficiently large $d_k$ and $d_{k+1}$,

$$I(T_{k+1}; T_k|w^*) \leq \frac{1}{2} \log \left( \frac{\prod_{i=1}^{d_{k+1}} \left( A_{ii} + \lambda_i + \frac{\sigma_z^2}{\sigma_{T_k}^2} \right)}{\prod_{i=1}^{d_{k+1}} \left( \lambda_i + \frac{\sigma_z^2}{\sigma_{T_k}^2} \right)} \right) = \tag{9}$$

$$\frac{1}{2} \sum_{i=1}^{d_{k+1}} \log \left( 1 + \frac{A_{ii}}{\lambda_i + \frac{\sigma_z^2}{\sigma_{T_k}^2}} \right) \xrightarrow[\sigma_z^2 \to 0]{} \frac{1}{2} \sum_{i=1}^{d_{k+1}} \log \left( 1 + \frac{A_{ii}}{\lambda_i} \right).$$

As previously established, $\delta w$ is a Brownian motion along the SGD iterations during the diffusion phase. This process is characterized by a low (and fixed) variance of the informative gradients (relevant dimensions), whereas the remaining irrelevant directions suffer from increasing variances as the diffusion proceeds (see, for example, (Sagun et al., 2017; Zhu et al., 2018; Jastrzebski et al., 2017)). In other words, we expect the "informative" $\lambda_i$ to remain fixed, while the irrelevant consistently grow, sub-linearly with time. Denote the set of "informative/relevant" directions as $\Lambda^*$ and the set of "non-informative" as $\Lambda_{NI}$. Then our final limit (9), as the number of SGD steps grow, is $I(T_{k+1}; T_k|w^*) \leq \frac{1}{2} \sum_{\lambda_i^* \in \Lambda^*} \log \left( 1 + \frac{A_{ii}}{\lambda_i^*} \right)$. Note that the directions that are compressed and the ones that are preserved depend on the required compression level. This is the reason that different layers converge to different values of $I(T_k; X)$.

### 4.3 Relation to other works

The analysis above suggests that the SGD compresses during the diffusion phase in many directions of the gradients. We argue that these directions are the ones in which the variance of the gradients is increasing (non-informative) whereas the information is preserved in the directions where the variance of the gradients remain small.

This statement is consistent with recent (independent) work on the statistical properties of gradients and generalization. Sagun et al. (2017); Zhu et al. (2018); Zhang et al. (2018b) showed that typically, the covariance matrix of the gradients is highly non-isotropic and that this is crucial for generalization by SGD. They suggested that the reason lies in the proximity of the gradients' covariance matrix to the Hessian of the loss approximation. Furthermore, it was argued by Zhang et al. (2018b); Keskar et al. (2016); Jastrzebski et al. (2017) that SGD tends to converge to flat minima. These flat minima often correspond to a better generalization. Zhang et al. (2018b) emphasized that SGD converges to flat minima values characterized by high entropy due to the non-isotropic nature of the gradients' covariance and its alignment with the error Hessian at the minima. In other words, all of the finding above suggest that good generalization performance is typically characterized by non-isotropic gradients and Hessian, that are in orthogonal directions to the flat minimum of the training error objective.

## 5 THE COMPUTATIONAL BENEFIT OF THE HIDDEN LAYERS

Our Gaussian bound on the representation compression (9) allows us to relate the convergence time of the layer representation information, $I(T_k; X)$, to the diffusion exponent $\alpha$, defined in section 2.3.

Denote the representation information at the diffusion time $\tau$ as $I(X; T_k)(\tau)$. It follows from (9) that

$$I(X; T_k)(\tau) \leq C + \frac{1}{2} \sum_{\lambda_i \in \Lambda^{NI}} \log \left( 1 + \frac{A_{ii}}{\lambda_i(\tau)} \right) \leq C + \frac{1}{2} \sum_{\lambda_i \in \Lambda^{NI}} \left( \frac{A_{ii}}{\lambda_i(\tau)} \right) \tag{10}$$

where $C$ depends on the informative information for this layer, but not on $\tau$.

Notice that $\lambda_i(\tau)$ are the singular values of the weights of a diffusion process, which grow as $\tau^\alpha$ where $\alpha$ is the diffusion exponent. Hence, $\lambda_i(\tau) = \lambda_i^0 \cdot \tau^\alpha$. Therefore, $I(X; T_k)(\tau) \leq C + \frac{1}{\tau^\alpha} \sum_{\lambda_i \in \Lambda^{NI}} \left( \frac{A_{ii}}{\lambda_i^0} \right)$.

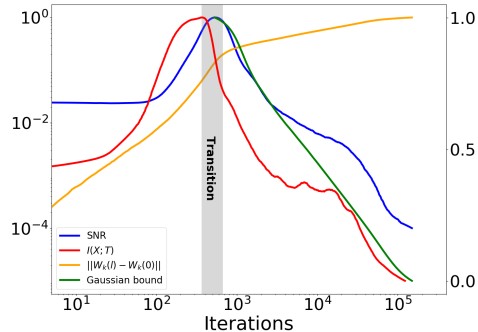
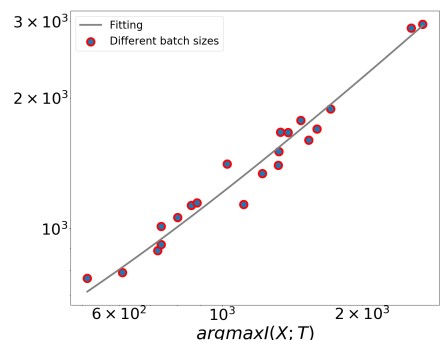

(a) The change of weights, the SNR of the gradients, the MI and the Gaussian bound during the training for one layer. In log-log scale

(b) The transition point of the SNR ($Y$-axis) versus the beginning of the information compression ($X$-axis), for different mini-batch sizes

Figure 1: MNIST data-set

Inverting this relation, the time to compress the representation $T_k$ by $\Delta I(X;T_k) = \Delta I_k$ scales as: $\tau(\Delta I_k) \propto \left(\frac{-R}{\Delta I(X;T)}\right)^{\frac{1}{\alpha}}$, where $R = \frac{1}{2}\sum_{\lambda_i \in \Lambda^{NI}}\left(\frac{A_{ii}}{\lambda_i^0}\right)$. Note that $R$ depends solely on the problem, $f(x)$ or $p(y,x)$, and not on the architecture. The idea behind this argument is as follows - one can expand the function in any orthogonal basis (e.g. Fourier transform). The expansion coefficients determine both the dimensionality of the relevant/informative dimensions and the total trace of the irrelevant directions. Since these traces are invariant to the specific function basis, these traces remain the same when expanding the function in the network functions using the weights.

Now, with $K$ hidden layers, where each layer only needs to compress from the previous (compressed) layer, by $\Delta I_k$ and the total compression is $\Delta I_X = \sum_k \Delta I_k$. Under these assumptions, even if the layers compress one after the other, the total compression time breaks down into $K$ smaller steps , as at $\left(\frac{R}{\sum_k \Delta I_k}\right)^{\frac{1}{\alpha}} \ll \sum_k \left(\frac{R}{\Delta I_k}\right)^{\frac{1}{\alpha}}$ if the $\Delta I_k$ are similar, we obtain a super-linear boost in the computational time by a factor $K^{\frac{1}{\alpha}}$. Since $\alpha \leq 0.5$ this is at least a quadratic boost in $K$. For ultra-slow diffusion we obtain an exponential boost (in $K$) in the convergence time to a good generalization. This is consistent with the observations reported by Shwartz-Ziv & Tishby (2017).

## 6 EXPERIMENTS

We now illustrate our results in a series of experiments. We examine several different setups.

**MNIST dataset**- In the first experiment, we evaluate the MNIST handwritten digit recognition task (LeCun et al., 1990). For this data set, we use a fully-connected network with 5 hidden layers of width $500 - 250 - 100 - 50 - 20$, with an hyperbolic tangent (tanh) activation function. The relative low dimension of the network and the bounded activation function allow us to empirically measure the MI in the network. The MI is estimated by binning the neurons' output into the interval $[-1, 1]$. The discretized values are then used to estimate the joint distributions and the corresponding MI, as described by Shwartz-Ziv & Tishby (2017).

Figure 1a depicts the norms of the weights, the signal-to-noise ratio (the ratio between the means of the gradients and their standard deviations), the compression rate $I(X;T)$ and the Gaussian upper bound on $I(X;T)$, as defined in (9). As expected, the two distinct phases correspond to the drift and diffusion phases. Further, these two phases are evident by independently observing the SNR, the change of the weights $||W(l) - W(0)||$, the MI and the upper bound. In the first phase, the weights grow almost linearly with the iterations, the SNR of the gradients is high, and there is almost no change in the MI. Then, after the transition point (that accrues almost at the same iteration for all

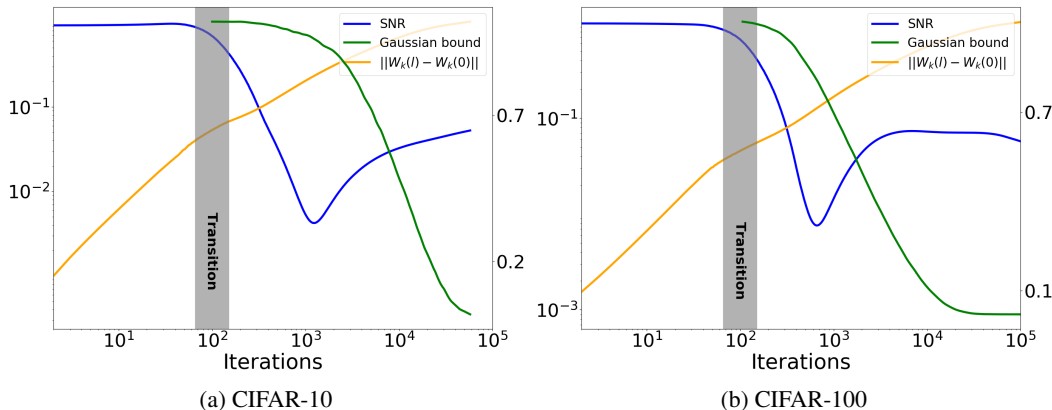

(a) CIFAR-10             (b) CIFAR-100

Figure 2: Change in the SNR of the gradients and the Gaussian bound on the MI during the training of the network for one layer on ResNet-32, in log-log scale.

the measures above), the weights behave as a diffusion process, and the SNR and the MI decrease remarkably. In this phase, there is also a clear-cut reduction of the bound.

**CIFAR-10 and CIFAR-100** - Next, we validate our theory on large-scale modern networks. In the second experiment we consider two large-scale data sets, CIFAR-10 and CIFAR-100. Here, we train a ResNet-32 network, using a standard architecture (including ReLU activation functions as described in (He et al., 2016). In this experiment we do not estimate the MI directly, due to the large scale of the problem. Figure 2 shows the SNR of the gradients and the Gaussian bound for one layer in CIFAR-10 and CIFAR-100 on the ResNet-32 network, averaged over 50 runs. Here, we observed similar behavior as reported in the MNIST experiment. Specifically, there is a clear distinction between the two phases and a reduction of the MI bound along the diffusion phase. Note that the same behavior was observed in most of the 32 layers in the network.

Recently there have been several attempts to characterize the correspondence between the diffusion rate of the SGD and the size of the mini-batch (Hu et al. (2017); Hoffer et al. (2017)). In these articles, the authors claimed that a larger mini-batch size corresponds to a lower diffusion rate. Here, we examined the effect of the mini-batch size on the transition phase in the Information Plane. For each mini-batch size, we found both the starting point of the information compression and the gradient phase transition (the iteration where the derivative of the SNR is maximal). Figure 1b illustrates the results. The $X$-axis is the iteration where the compression started, and the $Y$-axis is the iteration where the phase transition in the gradients accrued for different mini-batch sizes. There is a clear linear trend between the two. This further justifies our suggested model, since that the two measures are strongly related.

Next, we validate our results on the computational benefit of the layers. We train networks with a different number of layers (1-5 layers) and examine the iteration for which the network converge. Then, we find the $\alpha$ which fits the best trend $K^{\frac{1}{\alpha}}$ where $K$ is the number of layers. Figure 3 shows the results for two data-sets - MNIST and the symmetric dataset from Shwartz-Ziv & Tishby (2017). As our theory suggest, as we increase the number of layers, the convergence time decreases with a factor of $k^{\frac{1}{\alpha}}$ for different values of $\alpha$.

## 7 DISCUSSION AND CONCLUSIONS

In this work we study DNNs using information-theoretic principles. We describe the training process of the network as two separate phases, as has been previously done by others. In the first phase (drift) we show that $I(T_k; Y)$ increases, corresponding to improved generalization with ERM. In the second phase (diffusion), the representation information, $I(X; T_k)$ slowly decreases, while $I(T_K; Y)$ continues to increase. We rigorously prove that the representation compression is a direct consequence of the diffusion phase, independent of the non-linearity of the activation function.

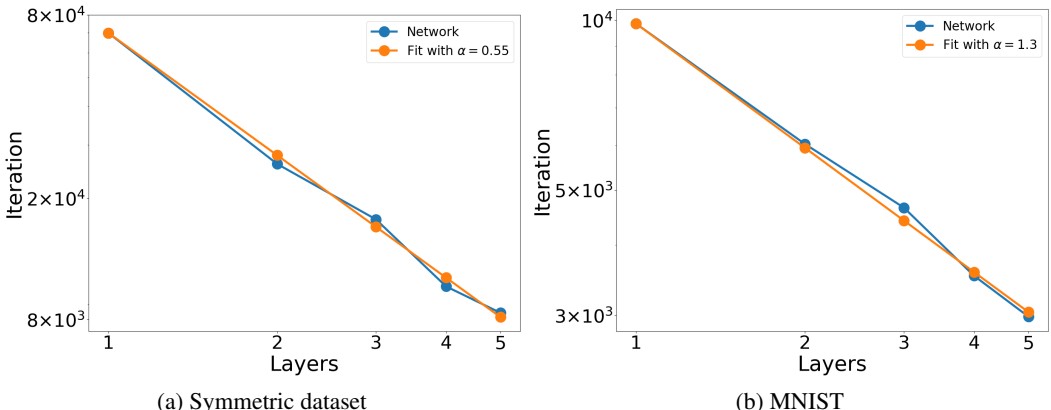

(a) Symmetric dataset          (b) MNIST

Figure 3: The computational benefit of the layers - The converged iteration as function of the number of layers in the network

We provide a new Gaussian bound on the representation compression and then relate the diffusion exponent to the compression time. One key outcome of this analysis is a novel proof of the computational benefit of the hidden layers, where we show that they boost the overall convergence time of the network by at least a factor of $K^2$, where $K$ is the number of non-degenerate hidden layers. This boost can be exponential in the number of hidden layers if the diffusion is "ultra slow", as recently reported.

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

## APPENDIX A - PROOF OF THEOREM 1

We first first revisit the well-known *Probably Approximately Correct* (PAC) bound. Let $\mathcal{H}$ be a finite set of hypotheses. Let $\ell_h(x_i, y_i)$ be a bounded loss function, for every $h \in \mathcal{H}$. For example, $\ell_h(x_i, y_i) = (y_i - h(x_i))^2$ is the squared loss while $\ell_h(x_i, y_i) = -y_i \log h(x_i)$ is the logarithmic loss (which may be treated as bounded, assuming that the underlying distribution is bounded away from zero and one). Let $\mathcal{L}_h(S_m) = \frac{1}{m}\sum_{i=1}^m \ell_h(x_i, y_i)$ be the empirical error. Hoeffding's inequality Hoeffding (1963) shows that for every $h \in \mathcal{H}$,

$$\mathbb{P}\left[\left|\mathcal{L}_h(S_m) - \mathbb{E}_{S_m}[\mathcal{L}_h(S_m)]\right| \geq \epsilon\right] \leq 2\exp\left(-2\epsilon^2 m\right). \tag{11}$$

Then, we can apply the union bound and conclude that

$$\mathbb{P}\left[\exists h \in \mathcal{H}\left|\left|\mathcal{L}_h(S_m) - \mathbb{E}_{S_m}[\mathcal{L}_h(S_m)]\right|\right| \geq \epsilon\right] \leq 2\left|\mathcal{H}\right|\exp\left(-2\epsilon^2 m\right).$$

We want to control the above probability with a confidence level of $\delta$. Therefore, we ask that $2\left|\mathcal{H}\right|\exp\left(-2\epsilon^2 m\right) \leq \delta$. This leads to a PAC bound, which states that for a fixed $m$ and for every $h \in \mathcal{H}$, we have with probability $1 - \delta$ that

$$\left|\mathcal{L}_h(S_n) - \mathbb{E}_{Sm}[\mathcal{L}_h(S_m)]\right|^2 \leq \frac{\log\left|\mathcal{H}\right| + \log\frac{2}{\delta}}{2m}. \tag{12}$$

Note that under the definitions stated in Section 1.1, we have that $\left|\mathcal{H}\right| \leq 2^{\mathcal{X}}$. However, the PAC bound above also holds for a infinite hypotheses class, where $\log\left|\mathcal{H}\right|$ is replaced with the VC dimension of the problem, with several additional constants (Vapnik & Chervonenkis, 1968; Shelah, 1972; Sauer, 1972).

Let us now assume that $X$ is a $d$-dimensional random vector which follows a Markov random field structure. As stated above, this means that $p(x_i) = \prod_i p(x_i|Pa(x_i))$ where $Pa(X_i)$ is a set of components in the vector $X$ that are adjacent to $X_i$. Assuming that the Markov random field is ergodic, we can define a *typical set* of realizations from $X$ as a set that satisfies the *Asymptotic Equipartition Property* (AEP) (Cover & Thomas, 2012). Therefore, for every $\epsilon > 0$, the probability of a sequence drawn from $X$ to be in the typical set $A_\epsilon$ is greater than $1 - \epsilon$ and $|A_\epsilon| \leq 2^{H(X)+\epsilon}$. Hence, if we only consider a typical realization of $X$ (as opposed to every possible realization), we have that asymptotically $|\mathcal{H}| \leq 2^{H(X)}$. Finally, let $T$ be a mapping of $X$. Then, $2^{H(X|T)}$ is the number of typical realizations of $X$ that are mapped to $T$. This means that the size of the typical set of $T$ is bounded from above by $2^{H(X)}/2^{H(X|T)} = 2^{I(X;T)}$. Plugging this into the PAC bound above yields that with probability $1 - \delta$, the typical squared generalization error of $T$, $\epsilon_T^2$ satisfies

$$\epsilon_T^2 \leq \frac{2^{I(X;T)} + \log\frac{2}{\delta}}{2m}. \tag{13}$$

## APPENDIX B - PROOF OF PROPOSITION 2

We make the following technical assumptions:

1. $w^*$ and $\delta w$ satisfy $\lim_{d_k \to \infty} w^{*T} \delta w = 0$ almost surely.

2. The moments of $T_k$ are finite.

3. The components of $w^*$ and $\delta w$ are *in-general-positions*, satisfying $\lim_{d_k \to \infty} \sum_{i=1}^{d_k} w_i^{*4} / \left( \sum_{i=1}^{d_k} w_i^{*2} \right)^2 = 0$ and $\lim_{d_k \to \infty} \sum_{i=1}^{d_k} \delta w_i^4 / \left( \sum_{i=1}^{d_k} \delta w_i^2 \right)^2 = 0$ almost surely.

Consider a sequence of i.i.d. random variable, $\{X_i\}_{i=1}^d$ with zero mean and finite moments, $\mathbb{E}[X_i^r] < \infty$ for every $r \geq 1$.

Let $\{a_i\}_{i=1}^d$ be a sequence of constants. Denote $Y_i = a_i X_i$, so that $\{Y_i\}_{i=1}^d$ are independent with zero mean and $\text{Var}(Y_i) = a_i^2 \mathbb{E}[X^2]$. Let $S = \sum_{i=1}^d a_i X_i = \sum_{i=1}^d Y_i$ and denote $U_d^2 = \sum_{i=1}^d \text{Var}(Y_i) = \mathbb{E}[X^2] \sum_{i=1}^d a_i^2$.

The Lyapunov Central Limit Theorem (CLT) Billingsley (2008) states that if there exists some $\delta > 0$ for which

$$\lim_{d \to \infty} \frac{1}{U_d^{2+\delta}} \sum_{i=1}^d \mathbb{E}\left[ |Y_i|^{2+\delta} \right] = 0 \tag{14}$$

then

$$\frac{1}{U_d} \sum_{i=1}^d Y_i \xrightarrow[d \to \infty]{\mathcal{D}} \mathcal{N}(0,1). \tag{15}$$

Plugging $\delta = 2$ yields the following sufficient condition,

$$\lim_{d \to \infty} \frac{1}{U_d^4} \sum_{i=1}^d \mathbb{E}\left[ Y_i^4 \right] = \frac{\sum_{i=1}^d a_i^4}{\left( \sum_{i=1}^d a_i^2 \right)^2} \frac{\mathbb{E}[X^4]}{\mathbb{E}^2[X^2]} = 0 \tag{16}$$

Let us apply the Lyapunov CLT to our problem. Here, the components of $T_k$ are i.i.d. for sufficiently large $d_k$, with zero mean and finite $r^{th}$ moments for every $r \geq 1$. Further, we assume that the components of $w^*$ and $\delta w$ are in-general-positions. This means that Lyapunov condition (16) is satisfied for both $w^{*T} T_k$ and $\delta w^T T_k$ almost surely, which means that

$$\frac{1}{\sqrt{\sigma_{T_k}^2} \|w^*\|_2} w^{*T} T_k \xrightarrow[d_k \to \infty]{\mathcal{D}} \mathcal{N}(0,1) \tag{17}$$

and

$$\frac{1}{\sqrt{\sigma_{T_k}^2}\|\delta w\|_2}\delta w^T T_k \xrightarrow[d_k\to\infty]{\mathcal{D}} \mathcal{N}(0,1). \tag{18}$$

almost surely, where $\sigma_{T_k}^2$ is the variance of the components of $T_k$.

Further, for every pair of constants $a$ and $b$, the linear combination $\left(aw^* + b\delta w\right)^T T_k$ also satisfies Lyapunov's condition almost surely, which means that $w^{*T}T_k$ and $\delta w^T T_k$ are asymptotically jointly Gaussian, with

$$\mathbb{E}\left[w^{*T}T_k \left(\delta w^T T_k\right)^T\right] = \sigma_{T_k}^2 w^{*T}\delta w \xrightarrow[d_k\to\infty]{} 0$$

almost surely. $\qquad\square$

