# OpenReview forum: "REPRESENTATION COMPRESSION AND GENERALIZATION IN DEEP NEURAL NETWORKS"
_ICLR.cc/2019/Conference_

### Official Review · AnonReviewer2 · 2018-11-02
**It is a paper written in a rush that its clarity is a main problem.**

**Rating:** 4
**Confidence:** 3

**Review:**

The authors are providing an information theoretic viewpoint on the behavior of DNN based on the information bottleneck.  The clarity of the paper is my main concern.  It contains quite a number of typos and errors.  For example, in section 6, the results of MNIST in the first experiment was presented after introducing the second experiment.  Also, the results shown in Fig 1b seems to have nothing to do with Fig. 1a.  It makes use of some existing results from other literature but it is not clearly explained how and why the results are being used.   It might be a very good paper if the writing could be improved.   The paper also contains some experimental results.  But they are too brief and I do not consider the experiments as sufficient to justify the correctness of the bounds proved in the paper.

---

### Official Review · AnonReviewer1 · 2018-11-05
**Similar to previous, fails to mention criticisms of the research program**

**Rating:** 3
**Confidence:** 3

**Review:**

This paper interprets the optimization of deep neural networks in terms of a two phase process: first a drift phase where gradients self average, and second a diffusion phase where the variance is larger than the square of the mean. As argued by first by Tishby and Zaslavsky and then by Shwartz-Ziv and Tishby (arxiv:1703.00810), the first phase corresponds to the hidden layers becoming more informative about the labels, and the second phase corresponds to a compression of the hidden representation keeping the informative content relatively fixed as in the information bottleneck of Tishby, Pereira, and Bialek.

A lot of this paper rehashes discussion from the prior work and does not seem sufficiently original. The main contribution seems to be a bound that is supposed to demonstrate representation compression in the diffusion phase. The authors further argue that this shows that adding hidden layers lead to a boosting of convergence time.

Furthermore, the analytic bound relies on a number of assumptions that make it difficult to evaluate. One example is using the continuum limit for SGD (1), which is very popular but not necessarily appropriate. (See, e.g., the discussion in section 2.3.3 in arxiv:1810.00004.)

Additionally, there has been extensive discussion in the literature regarding whether the results of Shwartz-Ziv and Tishby (arxiv:1703.00810) hold in general, centering in particular on whether there is a dependence on the choice of the hyperbolic tangent activation function. I find it highly problematic that the authors continue to do all their experiments using the hyperbolic tangent, even though they claim their analytic bounds are supposed to hold for any choice of activation. If the bound is general, why not include experimental results showing that claim? The lack of discussion of this point and the omission of such experiments is highly suspicious.

Perhaps more importantly, the authors do not even mention or address this contention or even cite this Saxe et al. paper (https://openreview.net/forum?id=ry_WPG-A-) that brings up this point. They also cite Gabrie et al. (arxiv:1805:09785) as promising work about computing mutual information for deep networks, while my interpretation of that work was pointing out that such methods are highly dependent on choices of binning or regulating continuous variables when computing mutual informations. In fact, I don't see any discussion at all this discretization problem, when it seems absolutely central to understanding whether there is a sensible interpretation of these results or not.

For all these reasons, I don't see how this paper can be published in its present form.

---

### Official Review · AnonReviewer3 · 2018-11-07
**Interesting, but hard to interpret the technical results.**

**Rating:** 6
**Confidence:** 3

**Review:**

This paper presents some results about the information bottleneck view of generalization in deep learning studied in recent work by Tishby et al.
Specifically this line of work seeks to understand the dynamics of stochastic gradient descent using information theory. In particular, it quantifies the mutual information between successive layers of a neural network. Minimizing mutual information subject to empirical accuracy intuitively corresponds to compression of the input and removal of superfluous information.
This paper further formalizes some of these intuitive ideas. In particular, it gives a variance/generalization bound in terms of mutual information and it proves an asymptotic upper bound on mutual information for the dynamics of SGD.

I think this is an intriguing line of work and this paper makes an meaningful contribution to it. The paper is generally well-written (modulo some typos), but it jumps into the technical details (stochastic calculus!) without giving much intuition to help digest the results or discussion of how they relate to the broader picture. (Although I appreciate the difficulty of working with a page limit.)

Typos, etc.:
p1. "ereas" should be "whereas"
p2. double comma preceeding "the weights are fixed realizations"
p5. extra of in "needed to represent of the data"
Thm 1. L(T_m) has not been formally defined when T_m contains a set of representations rather than data points.

---

### Author Response · Authors · 2018-11-16
**Authors' response to the reviewers' comments - part 1**

We thank the reviewers for their comments.

We agree with the reviewers that the submitted paper was not written carefully enough and requires major rewriting.

Yet, the reviewers, in particular reviewer 1, missed or dismissed our main and new results, which rigorously refutes - one by one - the misleading claims of Saxe et.al. [1].

The Information Bottleneck theory of Deep Learning [2-3] has received significant attention in the past year, as can be seen from the number of related, or inspired by, submissions to this conference alone. This is despite the fact that the theory was not properly and correctly described anywhere (certainly not by Saxe et al 2017 despite their title). Most of this impact is due to Tishby’s presentations and online talks. This is the reason we found it necessary to first review some of its basic claims. This review was obviously too long for this paper as the really new results were squeezed into the last pages.

Our main novel results are summarized below:

1. We provide a rigorous proof ( Thm. 2) that the mutual information between successive layers decreases during the diffusion phase of the SGD training - for any nonlinearity of the units, saturated, linear, or piecewise linear as ReLU.

2. The only important assumption in our proof is that there is a distinct diffusion phase in the SGD training, as reported and well established by many others [5-10]. This phenomenon is related to the convergence to “a flat minimum” of the training error. We also assume that the mini-batches are statistically independent and that the layers are sufficiently wide to justify our usage of the central limit theorem for the diffusion weights. All other assumptions are standard technical conditions which are met with probability 1 in standard deep learning. Our results do not rely in any way on continuous time SGD, nor on the assumption that the gradient fluctuations are Gaussian  - these requirements are clearly confusing and irrelevant. The continuous time approximation to SGD is in fact justified in [4], but is not essential to our analysis in this paper.

3. To demonstrate this result, numerical simulations in this paper have been done with ResNets with RelU nonlinearities, as explicitly stated in the paper - in contrast to the claim of reviewer 2.

---

> ### Author Response · Authors · 2018-11-16
> **Authors' response to the reviewers' comments - part 2**
>
> 4. This new bound directly leads to the empirical representation compression in the information plane, as reported by Shwartz-Ziv and Tishby for both saturated and ReLU nonlinearities, without any assumption on binning or discretization of the units!
>
> 5. This result refutes the main claims of Saxe et al: (a) that the observed compression depends on the binning (b) that it results from the saturation of the units and (c) has nothing to do with the stochastic gradients or generalization.
>
> 6. It also gives the first proof, to our knowledge, that convergence to flat minima improves generalization, as conjectured by many others without any mathematical explanation.
>
> 7. We finally briefly scratched (due to lack of space) our most striking corollary: due to this diffusion compression, the convergence to good generalization is faster with more hidden layers and the convergence time scales as a negative power of the number of effective layers.  We agree that this striking new result is hard to understand from this paper alone and requires a separate publication.
>
> References -
> [1]Saxe, A. M., Bansal, Y., Dapello, J., Advani, M., Kolchinsky, A., Tracey, B. D., & Cox, D. D. On the information bottleneck theory of deep learning, ICLR, 2018
> [2] Tishby, Naftali, and Noga Zaslavsky. "Deep learning and the information bottleneck principle." Information Theory Workshop (ITW), 2015 IEEE. IEEE, 2015.
> [3] Shwartz-Ziv, Ravid, and Naftali Tishby. "Opening the black box of deep neural networks via information." arXiv preprint arXiv:1703.00810 (2017).
> [4] Qianxiao Li, Cheng Tai, and Weinan E. Stochastic modified equations and adaptive stochastic gradient algorithms. arXiv:1511.06251, 2015.
> [5] Stephan Mandt, Matthew D Hoffman, and David M Blei. Stochastic gradient descent as approximate Bayesian inference. The Journal of Machine Learning Research, 18(1):4873–4907, 2017.
> [6] Chris Junchi Li, Lei Li, Junyang Qian, and Jian-Guo Liu. Batch size matters: A diffusion approximation framework on nonconvex stochastic gradient descent. arXiv:1705.07562v1, 2017
> [7] Samuel L Smith and Quoc V Le. A Bayesian perspective on generalization and stochastic gradient descent. arXiv:1710.06451, 2018.
> [5] Pratik Chaudhari and Stefano Soatto. Stochastic gradient descent performs variational inference, converges to limit cycles for deep networks. arXiv:1710.11029, 2017.
> [8] Stanislaw Jastrzebski, Zachary Kenton, Devansh Arpit, Nicolas Ballas, Asja Fischer, Yoshua Bengio, and Amos Storkey. Three factors influencing minima in SGD. arXiv:1711.04623, 2017.
> [9] Zhanxing Zhu, Jingfeng Wu, Bing Yu, Lei Wu, and Jinwen Ma. The anisotropic noise in stochastic gradient descent: Its behavior of escaping from minima and regularization effects. arXiv:1803.00195, 2018.
> [10] Jing An, Jianfeng Lu, and Lexing Ying. Stochastic modified equations for the asynchronous stochastic gradient descent. arXiv:1805.08244, 2018.

---

### Public Comment · (anonymous) · 2018-11-24
**related work**

The relation between compression (information reduction), flat minima (SGD), and generalization is also described in Achile https://arxiv.org/abs/1706.01350 which proves that flatness bounds information in the weights, and information in the weights bounds information in the activations, which is the form of compression discussed in this paper. That work should be referenced.

---

> ### Public Comment · (anonymous) · 2022-09-24
> **The claim of the previous work by Achile is invalid in the context of the presence paper**
>
>
>
> The above comment posted on 25 Nov 2018 is wrong. I will put a new comment here to correct the mistake.
>
> I reviewed the paper, Achile https://arxiv.org/abs/1706.01350, recommending for acceptance of the paper at JMLR. So, I feel obligated to avoid the error in the community here.
>
> The claim of Achile https://arxiv.org/abs/1706.01350 for the following is invalid: information in the weights bounds information in the activations, which is the form of compression discussed in this paper. Technically, Achile only shows that the negative entropy of the normal distribution bounds the information in the activations. Please see the proof of Proposition 4.1 to understand this. The negative entropy of the normal distribution and the information in the weight are very different, unless we make a strong and impractical assumption as is done in Achile. I still recommended for acceptance since this paper provides different other contributions than this one.
>
> Note that the normal distribution here is not even related to any distribution of the dataset or learning algorithm. Instead, the normal distribution is arbitrary, as it corresponds to the noise that authors added to the weights arbitrarily to get this result. So, we can change the entropy arbitrarily without changing the dataset distribution and learning algorithm. So, this cannot have any meaningful relation with information of weights and activations.
>
> But, if we say that "Achile https://arxiv.org/abs/1706.01350 proves that flatness bounds information in the weights, and information in the weights bounds information in the activations, which is the form of compression discussed in this paper", you are claiming that the negative entropy of the normal distribution (with an arbitrary variance) should be somehow a good approximation of the mutual information of weights and dataset and in turn a good bound on the mutual information of representation and input, in general or in practical settings. This is a strange claim and I think nobody will agree with this.
>
> So, for this aspect, Achile https://arxiv.org/abs/1706.01350 is using a bad presentation "trick" to hide the fact that their $\tilde I(w;S)$ is just the entropy of normal distribution H(b) (- a constant) where b is the normal random variable. So, first, $\tilde I(w;S)$ is not $I(w;S)$. Second, $\tilde I(w;S)$ is H(q) (- a constant). So, a better and more honest notation is to replace $\tilde I(w;S)$ with $\tilde H(q)$. Then, you can easily see that there is no technical contribution in this paper that connects information in the weights and information in the activations. But, again it is a good paper providing other contributions, which is why I recommended for acceptance.

---

### Meta-Review · Area_Chair1 · 2018-12-16
**Needs a rewrite**

**Confidence:** 4
**Recommendation:** Reject

**Metareview:**

The authors admit the paper "was not written carefully enough and requires major rewriting."  This seems to be a frustratingly common phenomenon with work on the information bottleneck.